# A detailed analysis of game statistics of professional tennis players: An inferential and machine learning approach

**Michal Bozděch** [1☯*], **Dominik Puda** [1☯], **Pavel Grasgruber** [2☯]

**1** Department of Physical Education and Social Sciences, Faculty of Sports Studies, Masaryk University, Brno, Czech Republic, **2** Department of Sport Performance and Exercise Testing, Faculty of Sports Studies, Masaryk University, Brno, Czech Republic

☯ These authors contributed equally to this work.
* michal.bozdech@fsps.muni.cz

## Abstract

Tennis, a widely enjoyed sport, motivates athletes and coaches to optimize training for competitive success. This retrospective predictive study examines anthropometric features and statistics of 1990 tennis players in the 2022 season, using 20,040 data points retrospectively obtained from the ATP official source after the end of the season. These data were cross-verified with information from other sources before categorisation to address any discrepancies. Employing various analytical methods, the results emphasize the strategic importance of tournament participation and gameplay for financial gains and higher rankings. Prize money analysis reveals a significant disparity favoring top players. Multivariate Analysis of Variance highlights the need to consider multiple variables for understanding ATP rankings. Multinomial Logistic Regression identifies age, height, and specific service-related metrics as key determinants, with older and taller players more likely to secure top positions. Neural Network models exhibit potential in predicting ATP Rank outcomes, particularly for ATP Rank (500). Our results argue for the use of Artificial Intelligence (AI), specifically Neural Networks, in handling complex interactions and emphasize that AI is a supportive tool in decision-making, requiring careful consideration by experienced individuals. In summary, this study enhances our understanding of ATP ranking factors, providing actionable insights for coaches, players, and stakeholders in the tennis community.

## Introduction

Analyzing one's own as well as the opponent's game statistics is a crucial aspect of player preparation. This preparation also aids in formulating a strategy and tactics based on one's strengths and the opponent's weaknesses, enabling the tennis player to better anticipate the opponent's moves and performance during the game [1, 2]. Consequently, it is possible to reduce the negative outcomes of performance that could be caused by one's own shortcomings or the actions of the opponent [3–5]. For these reasons, coaches and players analyze physical fitness, technical and psychical factors, good and bad performances of players according to

**Data Availability Statement:** The dataset supporting the study results can be found on figshare at https://doi.org/10.6084/m9.figshare.24763872.v1. The NN code of the best-performing model can be found on figshare at https://doi.org/10.6084/m9.figshare.24762705.v1.

**Funding:** The author(s) received no specific funding for this work.

**Competing interests:** The authors have declared that no competing interests exist.

performance indicators that include biomechanics, successful game patterns, physiological and psychological techniques [6–9]. These factors are used to distinguish the successful from the less successful players in a single game, tournament, and season, which makes them particularly relevant for training purposes throughout the season. These analyzes are time-, expert-, and financial-intensive, and only selected individuals can afford them.

There are five main goals in tennis performance analysis, also known as notational analysis: assessing tactics, evaluating technique, analyzing movement, building a database and models, and providing educational tools for both coaches and players [10]. These objectives, initially proposed by Hughes [10], were further developed by O'Donoghue [11]. He emphasized the future prospects of transforming match analysis through advanced techniques and the importance of practical match analysis within coaching contexts. The development of analytical techniques was previously discussed by Liebermann et al. [12]. When describing analytical methods for sports performance using the latest IT technology at the time, Liebermann et al. [12] proposed that these technologies should be utilized in everyday coaching. In relation to this definition, it can be stated that the mentioned methods are fully applicable today, and thus coaches, athletes, and stakeholders now have at their disposal more modern methods for obtaining, analyzing, and interpreting results than those to which the above definitions were related. One of these advanced methods is Big Data analytics using Artificial Intelligence (AI), more precisely Machine Learning [13]. These advanced AI methods have been utilized in various ways within tennis. For example, it has been found that tennis strokes can be classified using data from a personal wristband [14]. Additionally, predictions of match winners can be made before the start of a match by analysing data from players' previous performances, capturing their short, medium, and long-term performances, as well as their affinity for different types of surfaces [15]. More precisely, the success depends on the serve direction (serves directed more than 5.88° away from the receiving player have a higher chance of ace), the proximity of the ball's landing spot to the nearest service box line (a ball landing less than 15.27 cm away from the service box line has a greater chance of becoming an ace), and the serve speed. Although serve speed is less significant than accuracy, it still plays a crucial role in matches [16]. Our study aims to process a relatively large and specific volume of data from which it will be possible to predict practically applicable and significant factors. These factors could then be used by coaches, athletes and stakeholders, saving them from having to go through the time-consuming data analysis and decision-making process themselves.

As was evident from the previous information, we can monitor and analyze many variables in tennis. One of the most important variables is Serve. The significance of serving stems from the fact that it constitutes the first action in the game. Subsequently, each return is dependent on this initial action. These interactions often serve as indicators for specific technical and conditioning training programs [17], especially the serve velocity [18]. Tennis return, but most importantly serve are the two most essential strokes in tennis, and their level improves with professional ranking [19–22]. An increasing serve speed reduces the time for the opponent to return the ball successfully and increases the probability of the server's superiority in the following game or of gaining a direct point [16, 20, 23]. Therefore, we also expect a relatively greater importance for this variable.

There are many ways and methods to acquire, process and analyze data [13]. Data collection methods encompass the use of specially designed computer systems, mathematical models, and video analysis [24–30]. The studies address a wide range of tennis match aspects, including player statistics, technical characteristics, match durations, and tactical strategies. Some studies concentrate on predicting tennis match outcomes through mathematical models and player performance analysis [31–41]. These studies emphasize the importance of detailed data, such as serving success, return success, probabilities of rare events, and other specific

match statistics [27, 42–49]. Several studies explore relationships between players' physical characteristics, such as height, and their performance, e.g. serve speed [23, 50]. Some studies have looked at long-term trends and changes in players' performance over the years and the impact of age on their rankings [51, 52]. Therefore, performance analysis is a very wide term with a wide range of combinable options for a detailed and practically applicable analysis. However, we have chosen a procedure (using publicly available secondary data) that would enable coaches, athletes and stakeholders to make better decisions about strategies and tactics. On selected and important data that emerged from the findings of this study.

It is important to note, that tennis is significantly influenced by rapidly evolving trends, including the increasing speed of serves [34]. Modern tennis stands out for its high dynamism, speed in thinking and action, precision, and high technical and tactical skills [49, 53]. Therefore, the conclusions drawn from our study are not definitive and may evolve over time. For accuracy, it is essential to base decisions on these findings and individualise them rather than generalise. The authors emphasize the importance of the return of serve as one of the key shots in tennis, along with the serve, as the first action in the game. They have shown that the player with the highest percentage of successful service returns is the one who wins the match, once again highlighting its significance. Even on slow surfaces like clay [27], serves and service returns remain the strokes that most influence match outcomes in modern tennis [54]. This gameplay dynamics and the pivotal role of serves and service returns underscore the current trajectory of tennis and therefore had a significant role in the preparation of the research plan for this project.

This study aims to analyse the performance (service, return) and anthropometric characteristics of professional tennis players to define and quantify the significance of the main factors influencing rankings, points, and prize money at the end of the season. The findings of this study can help coaches, players, and stakeholders better understand these factors. Moreover, by comparing individual player statistics, the existing training plans can be updated to enhance the likelihood of achieving a better ranking at the conclusion of the current tennis season.

## Materials and methods

To achieve our research objectives, we conducted meticulous data collection and analysis. The data used to verify our retrospective research premises were sourced from the official website of men's professional tennis, specifically https://atptour.com. Due to publicly available access to the data, all ATP-registered tennis players were required to sign informed consent to share personal data. After the culmination of the 2022 player season, data regarding player characteristics were acquired in July 2023. This data is maintained, updated, and monitored by the Association of Tennis Professionals (ATP). During data collection, no author had access to the entire file. After the conclusion of data collection, the data were anonymized. Our analysis focused exclusively on singles tennis matches, excluding doubles. The acquired data underwent a rigorous cleansing process, which involved the removal of typographical errors and outliers. Outliers were individually identified through cross-referencing with other official online sources, and values were either revised based on verified evidence or removed from the database altogether. Subsequently, these cleansed data were classified according to our research criteria and evaluated in alignment with our research objectives.

In this study, a total of 31 independent and dependent variables were processed, which were both ordinal and continuous in nature. The data from all 1990 ATP tennis players pertaining to the 2022 season were acquired after its conclusion. Data collection involved the participation of all authors, and the principal author subsequently performed data quality control and corrections. The research encompassed all registered tennis players in the ATP rankings for the 2022 season. No player was excluded, except for individual unrealistic values.

## Study criteria

**ATP rank.** Based on players' end-of-season rankings, the tennis players were divided into intervals below 100 (ATP (100)), 300 (ATP (300)), and 500 (ATP (500)). For instance, a player ranked 604th was categorized into 7, 3, and 2, respectively. Players with lower rankings were merged into a single research category due to the low prevalence of data, especially in the case of player game statistics (see Table 1). More precisely, ATP (100) consisted of 10 independent categories, with rankings 1 to 9 falling within intervals of 100 ranks in each category, and the 10th category including players ranked beyond 900. ATP (300) consisted of 4 independent categories, with rankings 1 to 3 falling within intervals of 300 ranks in each category, and the 4th category including players ranked beyond 900. ATP (500) consisted of 3 independent categories, with rankings 1 to 2 falling within intervals of 500 ranks in each category, and the 3rd category including players ranked beyond 1000.

**Anthropometric characteristics.** For the sake of clarity, we included variables Age (in years), Height (in cm), and Weight (in kg) in this category.

**Serve.** This category encompassed 10 player statistics from the single service record, specifically: Aces (sum of integers), Double Faults (sum of integers), 1st Serve (percentage), 1st Serve Points Won (percentage), 2nd Serve Points Won (percentage), Break Points Faced (sum

**Table 1. Basic descriptive statistic of all registered ATP players from 2022 season.**

| Variable | n | Mean | SD | Min | Max | $\gamma_1$ | $\gamma_2$ | Med ($x_{25}$ _$x_{75}$) |
|---|---|---|---|---|---|---|---|---|
| Age, year | 1978 | 23.70 | 4.48 | 15.00 | 44.00 | 0.86 | 0.59 | 23.00 (20.00–26.00) |
| Points, n | 1989 | 123.82 | 428.18 | 1.00 | 6820.00 | 8.81 | 100.20 | 11.00 (2.00–72.00) |
| Tournament Played, n | 1989 | 12.54 | 8.85 | 1.00 | 38.00 | 0.43 | -1.05 | 11.00 (5.00–20.00) |
| Weight, kg | 1399 | 77.99 | 6.79 | 56.00 | 110.00 | 0.22 | 0.66 | 78.00 (73.00–82.00) |
| Height, cm | 1379 | 184.06 | 6.86 | 132.00 | 211.00 | -0.19 | 2.38 | 183.00 (180.00–188.00) |
| Win, n | 1952 | 1.53 | 6.33 | 0.00 | 61.00 | 5.48 | 33.18 | 0.00 (0.00–0.00) |
| Lose, n | 1952 | 1.50 | 4.89 | 0.00 | 30.00 | 3.94 | 15.19 | 0.00 (0.00–0.00) |
| Prize money, $ | 1931 | 105442.49 | 509849.09 | 0.00 | 9934581 | 10.83 | 151.76 | 4222.00 (1527.00–17534.00) |
| Aces, n | 304 | 106.75 | 162.26 | 0.00 | 895.00 | 2.48 | 6.93 | 34.00 (6.00–148.5) |
| Double Faults, n | 304 | 50.69 | 65.52 | 0.00 | 439.00 | 2.14 | 6.33 | 18.00 (6.00–76.00) |
| 1st Serve, % | 304 | 0.62 | 0.05 | 0.41 | 0.80 | -0.37 | 1.08 | 0.63 (0.59–0.66) |
| 1st Serve Points Won, % | 304 | 0.69 | 0.07 | 0.45 | 0.91 | -0.61 | 1.04 | 0.69 (0.65–0.73) |
| 2nd Serve Points Won, % | 304 | 0.48 | 0.07 | 0.18 | 0.64 | -1.15 | 2.63 | 0.49 (0.45–0.52) |
| Break Points Faced, n | 304 | 119.86 | 133.21 | 1.00 | 602.00 | 1.09 | 0.08 | 51.00 (14.00–205.75) |
| Break Points Saved, % | 304 | 0.57 | 0.13 | 0.00 | 0.79 | -2.19 | 6.45 | 0.60 (0.53–0.63) |
| Service Games Played, n | 304 | 215.54 | 251.97 | 1.02 | 912.00 | 1.13 | 0.02 | 85.00 (19.00–362.75) |
| Service Games Won, % | 304 | 0.73 | 0.13 | 0.14 | 0.93 | -1.96 | 5.91 | 0.75 (0.68–0.81) |
| Total Service Points Won, % | 304 | 0.61 | 0.06 | 0.37 | 0.74 | -1.00 | 2.04 | 0.61 (0.58–0.64) |
| 1st Serve Return Points Won, % | 304 | 0.27 | 0.06 | 0.04 | 0.43 | -0.94 | 2.04 | 0.27 (0.24–0.30) |
| 2nd Serve Return Points Won, % | 304 | 0.47 | 0.07 | 0.22 | 0.65 | -0.84 | 1.78 | 0.48 (0.44–0.51) |
| Break Points Oppor., n | 304 | 119.08 | 151.78 | 0.00 | 684.00 | 1.40 | 0.96 | 40.50 (9.75–184) |
| Break Points Conver., % | 304 | 0.36 | 0.17 | 0.00 | 1.00 | 0.45 | 3.36 | 0.38 (0.31–0.42) |
| Return Games Played, n | 304 | 218.98 | 255.36 | 1.05 | 996.00 | 1.14 | 0.07 | 87.00 (20.50–364.25) |
| Return Games Won, % | 304 | 0.17 | 0.08 | 0.00 | 0.43 | -0.25 | 0.33 | 0.18 (0.13–0.22) |
| Return Points Won, n | 304 | 0.34 | 0.05 | 0.16 | 0.47 | -1.01 | 1.55 | 0.35 (0.32–0.37) |
| Total Points Won, n | 304 | 0.48 | 0.04 | 0.33 | 0.55 | -1.47 | 2.76 | 0.48 (0.46–0.50) |

$\gamma_1$, Skewness; $\gamma_2$, Kurtosis; $x_{25}$, 25th percentile; Med ($x_{50}$), median (50th percentile); $x_{75}$, 75th percentile.

of integers), Break Points Saved (percentage), Service Games Played (sum of integers), Service Games Won (percentage), Total Service Points Won (sum of integers).

**Serve (%).** This subset included all percentage variables from the Serve category.

**Return.** This category encompassed eight player statistics from the single return record, specifically: 1st Serve Return Points Won (percentage), 2nd Serve Return Points Won (percentage), Break Points Opportunities (sum of integers), Break Points Converted (percentage), Return Games Played (sum of integers), Return Games Won (percentage), Return Points Won (percentage), Total Points Won (percentage).

**Return (%).** This subset included all percentages variables from the Return category.

**Service, Return.** This subset included all integer variables from the Service and Return categories.

**Service, Return (%).** This subset included all percentage variables from the Service and Return categories.

Variables Prize money (in dollars), Points (sum of integers), and Tournaments Played (sum of integers) were used only in descriptive statistics or predictive Neural Network models due to their high multicollinearity with independent variables (ATP Ranks).

## Statistical analysis

**Utilizing the multivariate analysis of variance.** Multivariate Analysis of Variance (MANOVA) was used as a powerful statistical technique employed to investigate the impact of one or more independent variables (IVs) on multiple dependent variables (DVs) simultaneously. Within this study, the primary objective was to assess the influence of different levels of ATP ranks (Fixed factor or independent variable, IV), specifically ATP (100), ATP (300), and ATP (500), on a set of DVs. These DVs were categorized into three groups: Anthropometric measurements, Service-related metrics, and Return-related metrics. The characteristics of these grouped IVs and DVs have been described in detail above. Notably, only DVs exhibiting a ratio or percentage scale were considered in our analysis, excluding integer-based variables. Prior to conducting MANOVA, the data were screened for missing values, outliers, and adherence to the scale requirements (ratio/percentage). Given the inherent assumptions associated with MANOVA, it is crucial to note that our data did not meet these assumptions fully. Specifically, the assumptions of multivariate normality, homogeneity of variance-covariance matrices, and linearity were violated. To address these violations, we utilized Pillai's Trace as an alternative test statistic, which is robust to these assumption violations. To gain further insights into the intergroup differences resulting from the various ATP levels, we employed a combination of post hoc tests. Specifically, we utilized Scheffe's Post Hoc test (selected for its flexibility and robustness in handling multiple comparisons), and Bonferroni correction (applied to control for Type I error across multiple pairwise comparisons). Furthermore, to gauge the magnitude of the observed effects, we employed Partial eta squared ($\eta^2_p$) as an effect size measure. The thresholds for interpreting effect sizes, as defined by Cohen [55], were categorized as follows: small ($\eta^2_p = 0.01$), medium ($\eta^2_p = 0.06$), and large ($\eta^2_p = 0.14$) effects.

**Utilizing the Multinomial Logistic Regression.** In this section, we will discuss the application of Multinomial Logistic Regression (MLR) as part of our statistical analysis. MLR was employed to predict categorical outcomes when dealing exclusively with metric data. We will elucidate the predictors and explanatory variables used in our models, along with the process of model evaluation and assumption verification.

Our predictive model centres around the following metric variables, which are used to predict categorical outcomes: ATP (100), ATP (300) and ATP (500). The explanatory variables considered in our analysis encompass the following factors: Anthropometric measurements,

Service-related metrics, and Return-related metrics. To enhance the relevance and interpretability of our findings, we incorporated reference data, specifically, the categories (ranks) of the lowest-scoring players in the ATP ranking system.

To ascertain whether the full model demonstrates a substantial improvement in fit over the null model, model fitting information was rigorously evaluated. Statistically significant results ($p < .001$) were obtained for all nine models, each encompassing three types of IV (ATP levels: ATP (100), ATP (300), and ATP (500)) and three types of DV (Anthropometric, Service, Return). A goodness-of-fit assessment was conducted to determine whether the model fits the observed data. Both the Pearson and Deviance tests yielded statistically non-significant results, except for the Deviance test in the case of ATP (500) and the anthropometric model ($\chi^2(2030)$ = 2270.55, $p < .001$). Since the Pearson test did not reach statistical significance ($\chi^2(2030)$ = 2045.48, $p = .400$), we retained the model.

Subsequently, in the results, we include information about the Likelihood Ratio tests used to assess the overall contribution of each IV to the model. Variables that were statistically significant predictors in the model were determined using the conventional alpha level of 0.05. Furthermore, the Parameter estimates, with the Bonferroni correction, was applied.

**Utilizing the artificial intelligence.** In this section, we explore the application of artificial intelligence within the context of our research. We outline the research criteria, including IVs, predictors, and the architecture of our models, all of which utilize artificial intelligence techniques. The file with exported AI code with synaptic weight estimates, activation function et cetera of the best-performing Neural Network model is available on the public repository; named NN 3.1 model [56].

*Research criteria.* Our analysis encompasses five independent response variables: ATP (100), ATP (300), ATP (500), Prize money, Points. Predictors in our models encompass various combinations, including the aggregation of all predictors, service-related variables, return-related variables, and combinations of Service with Return as well as Service (%) with Return (%). Additionally, the All model, in contrast to the others, incorporates anthropometric (Age, Weight, Height) and additional factor Tournament Played.

*Input and output layers.* The input and output layers of our models vary in terms of the number of units and the types of variables they encompass. Specifically, in the case of models incorporating all predictors, the input layer consists of 22 units accommodating a comprehensive set of predictors, including anthropometric measurements (Age, Height, Weight), service-related variables (Service, Service (%)), and return-related variables (Return, Return (%)). Meanwhile, the output layer for categorical response variables, such as ATP (100), ATP (300), and ATP (500), comprises a specific number of units corresponding to the categories within each variable. For continuous response variables like Prize money and Points, the output layer consists of continuous (scale) variables, allowing for regression-style predictions. These input and output layer configurations are consistent across all models, facilitating the application of artificial intelligence techniques for predictive modeling and analysis.

*Data splitting.* To ensure consistency and comparability, all models applied were of the Neural Network type (NN), specifically multilayer perceptron. Cases were randomly assigned based on the relative number of cases, with 70% allocated to training and 30% to testing. The model architecture comprised two hidden layers, employing hyperbolic tangent as the activation function and Identity as the output activation function. Batch training and the Scaled Conjugate Optimization algorithm were utilized.

*Receiver Operating Characteristics analysis.* The Receiver Operating Characteristics (ROC) curve illustrates the sensitivity against 1-specificity. The Area Under the Curve (AUC) is an interpretation of this probability and serves as an indicator of the overall model quality when dealing solely with categorical responses. While there is no universally accepted threshold

classification paradigm, in our research, we utilized the following interpretations: fail (.50-.60), poor (.60-.70), fair (.70-.80), good (.80-.90), and excellent (.90–1.00) [57].

## Software utilization

Given the extensive and intricate nature of our data analysis and the imperative need for rigorous data evaluation and result control, we utilized the capabilities of three distinct statistical software programs. Specifically, IBM SPSS Statistics (IBM Corp, v. 29.0.0.0) played a central role in our data analysis process, serving as the linchpin for critical calculations, including MANOVA, MLR, NN modelling and generating graphs. Complementing SPSS, we leveraged the open-source software JASP (v. 0.17.2.1) for initial descriptive statistical analyses. Additionally, MATLAB, equipped with machine learning toolboxes (MathWorks, R2022a), contributed significantly to our artificial intelligence-based modeling and analysis. This multifaceted software approach ensured the comprehensive and reliable examination of our data. JASP was employed for initial descriptive statistical analyses and nonparametric correlation (Spearman's rho with 95% confidence interval) to provide a fundamental overview of our dataset, while MATLAB was instrumental in implementing artificial intelligence techniques, notably Cross-Validation with a 5-fold approach. The combination of these statistical software tools ensured the comprehensive and reliable analysis of our data.

## Ethical considerations

As the data used in this study were publicly available, there was no need to seek ethical approval from a review board. While our study did not directly involve human subjects, we upheld ethical standards in the analysis of secondary data. We followed established data ethics principles, ensuring data privacy, confidentiality, and responsible data use. Although primarily designed for research with human participants, the Helsinki Declaration's ethical guidelines served as a reference point for our broader ethical framework. These guidelines emphasize privacy, data security, and research integrity, principles we incorporated into our data analysis to maintain ethical standards even in a non-human subject context.

## Results

In Table 1, the results of descriptive statistics for all 26 research variables can be seen. The Win and Lose variables will not be further examined in this study. From the table, it is evident that the youngest player was 15 years old. There was a total of 5 players ranked from 1373rd to 1726th who participated in 1–3 tournaments, earning 1–3 points and thereby making $258-$1024. Conversely, the oldest player (Toshihide Matsui) was 44 years old, achieved a rank of 1517, played in 3 tournaments, earned 2 points, and made $2015.

We can thus conclude that there is a certain parallel between these extreme examples. The highest-ranked player (Carlos Alcaraz) has earned the most points despite only participating in 17 tournaments, while two players with ranks 572 and 872 participated in 38 tournaments, earning 17–55 points and $8044-$15707. Hence, we were interested in the association between rank and the ratio of points earned per tournament played. A very strong and negative association was found (rho = -.912; p < .001; 95% CI = -.919, -.904). The relationship between rank and tournaments played alone exhibited a lower level of association (rho = -.831; p < .001; 95% CI = -.844, -.816), as did the positive association between the number of tournaments played and points earned (rho = .853; p < .001; 95% CI = .841, .865). Therefore, we can conclude that the strategy of gaining a minimum number of points over multiple tournaments (instead of thorough preparation) might be suitable for gaining experience but, according to the data, is not conducive to achieving a higher ranking in the ATP. It should be noted that the

**Table 2. Summary results of multivariate test for MANOVA.**

| IV | DV | Value | F | Hyp. df | Error df | p | $\eta^2_P$ | NCP | 1-β |
|---|---|---|---|---|---|---|---|---|---|
| ATP (100) | Anthropometric | 0.08 | 4.16 | 27 | 4098 | < .001 | .027* | 112.379 | 1.00 |
| | Service (%) | 0.75 | 4.68 | 54 | 1764 | < .001 | .125** | 252.825 | 1.00 |
| | Return (%) | 0.71 | 4.36 | 54 | 1764 | < .001 | .118** | 235.497 | 1.00 |
| ATP (300) | Anthropometric | 0.05 | 8.36 | 9 | 4116 | < .001 | .018* | 75.249 | 1.00 |
| | Service (%) | 0.46 | 8.93 | 18 | 891 | < .001 | .153*** | 160.700 | 1.00 |
| | Return (%) | 0.31 | 5.79 | 18 | 891 | < .001 | .105** | 104.239 | 1.00 |
| ATP (500) | Anthropometric | 0.06 | 13.11 | 6 | 2744 | < .001 | .028* | 78.664 | 1.00 |
| | Service (%) | .36 | 10,75 | 12 | 594 | < .001 | .178*** | 128.953 | 1.00 |
| | Return (%) | 0.18 | 5.03 | 12 | 594 | < .001 | .092** | 60.342 | 1.00 |

$\eta^2_P$, Partial Eta Squared; NCP, Non-Centrality Parameter; 1-β, Observed Power.

*small effect size; **medium effect size; ***large effect size.

association between rank and points was the highest among these variables (rho = -.9971, p < .001; 95% CI = -.9973, -.9967). This conclusion is logical and stems from the fact that Rank in tennis is formed using Points.

Furthermore, it would be logical to assume that the top player (Carlos Alcaraz) earned the most Prize money. However, even though he ranked second (in terms of Prize money earned), he earned $2,306,969 (23.22%) less than Novak Djokovic. The disparity in prize money between these two highest values is evident throughout the dataset, as confirmed by the Skewness ($\gamma_1$) and Kurtosis ($\gamma_2$) findings, which reached relatively higher values compared to other observed variables. Specifically, the Skewness confirmed a strong positive (right-skewed) asymmetry in the data distribution, indicating that the majority of research data falls below the mean. Using the Kurtosis coefficient, it can be stated that the frequency distribution is more peaked than a normal distribution (known as Leptokurtic). From these conclusions, it can be inferred that most tennis players in the 2022 season earned less Prize money than the stated $105,442.49±509,849.09. More precisely, out of a total of 1931 valid values of prize money, 237 tennis players (12.27%) received more than the average prize money, while 1694 players (87.73%) from the 2022 season received less than the average prize money.

In the following subsections, these data on DVs will be further analyzed and compared according to the predetermined IVs for the research.

## Differences in ATP rank effects on anthropometric, service, and return metrics

We used MANOVA as an extension of the univariate analysis of variance for examining the group differences of a singular IV across multiple outcome variables.

From Table 2, it is evident that a statistically significant difference exists across the levels of the IV in a linear combination of the DV. As a result, it is appropriate to proceed with a detailed statistical analysis. Furthermore, since Partial Eta Squared indicates the variance in the DV that can be explained by the IV, we can infer that the DV can explain variations ranging from small ($\eta^2_p$ = .027) to large ($\eta^2_p$ = .178) in ATP status (the IV). In other words, considering the example of the strongest effect, we can conclude that 17.8% of the variance in the variable Service (%) can be attributed to ATP (500). Unlike p-values, it is permissible to compare effect size values among themselves. Therefore, it is worth noting that the variable Service (%) also consistently explained the largest variation within each IV; ATP (100), ATP (300) and ATP (500). These are logical conclusions that indicate the game begins with the serve; hence,

**Table 3. Summary of the univariate tests results for different MANOVAs.**

| IV | DV | Sum of Squares | df | Mean Square | F | p | $\eta^2_P$ | NCP | 1-β |
|---|---|---|---|---|---|---|---|---|---|
| Rank100 | Age | 1487.78 | 9 | 165.31 | 8.26 | < .001 | .052 | 74.35 | 1.00 |
| | Weight | 1016.98 | 9 | 113.00 | 2.40 | .011 | .016 | 21.55 | .93 |
| | Height | 1925.75 | 9 | 213.97 | 4.67 | < .001 | .030 | 42.04 | 1.00 |
| Rank 300 | Age | 1151.10 | 3 | 383.70 | 19.03 | < .001 | .040 | 57.08 | 1.00 |
| | Weight | 591.39 | 3 | 197.13 | 4.17 | .006 | .009 | 12.51 | .86 |
| | Height | 1000.08 | 3 | 333.36 | 7.20 | < .001 | .016 | 21.61 | .98 |
| Rank 500 | Age | 1160.17 | 2 | 580.09 | 28.79 | < .001 | .040 | 57.59 | 1.00 |
| | Weight | 702.85 | 2 | 351.42 | 7.45 | < .001 | .011 | 14.90 | .94 |
| | Height | 1136.86 | 2 | 568.43 | 12.32 | < .001 | .018 | 24.64 | 1.00 |
| Rank 100 | 1st Serve | 0.04 | 9 | 0.01 | 1.63 | .107 | .047 | 14.64 | .75 |
| | 1st Serve Points Won | 0.24 | 9 | 0.03 | 6.17 | < .001 | .159 | 55.53 | 1.00 |
| | 2nd Serve Points Won | 0.29 | 9 | 0.03 | 9.19 | < .001 | .220 | 82.75 | 1.00 |
| | Break Points Saved | 261.03 | 9 | 29.00 | 7.47 | < .001 | .186 | 67.26 | 1.00 |
| | Service Games Won | 1.33 | 9 | 0.15 | 12.78 | < .001 | .281 | 115.05 | 1.00 |
| | Total Service Points Won | 0.25 | 9 | 0.03 | 10.94 | < .001 | .251 | 98.42 | 1.00 |
| Rank 300 | 1st Serve | 0.03 | 3 | 0.01 | 3.44 | .017 | .033 | 10.33 | .77 |
| | 1st Serve Points Won | 0.17 | 3 | 0.06 | 12.30 | < .001 | .110 | 36.90 | 1.00 |
| | 2nd Serve Points Won | 0.13 | 3 | 0.05 | 11.58 | < .001 | .104 | 34.73 | 1.00 |
| | Break Points Saved | 260.84 | 3 | 86.95 | 22.86 | < .001 | .186 | 68.57 | 1.00 |
| | Service Games Won | 1.01 | 3 | 0.34 | 27.14 | < .001 | .213 | 81.43 | 1.00 |
| | Total Service Points Won | 0.16 | 3 | 0.06 | 19.81 | < .001 | .165 | 59.43 | 1.00 |
| Rank 500 | 1st Serve | 0.01 | 2 | 0.01 | 1.88 | .155 | .012 | 3.76 | .39 |
| | 1st Serve Points Won | 0.13 | 2 | 0.06 | 13.73 | < .001 | .084 | 27.45 | 1.00 |
| | 2nd Serve Points Won | 0.08 | 2 | 0.04 | 9.80 | < .001 | .061 | 19.59 | .98 |
| | Break Points Saved | 327.70 | 2 | 163.85 | 45.91 | < .001 | .234 | 91.82 | 1.00 |
| | Service Games Won | 0.56 | 2 | 0.28 | 20.11 | < .001 | .118 | 40.21 | 1.00 |
| | Total Service Points Won | 0.11 | 2 | 0.06 | 18.57 | < .001 | .110 | 37.13 | 1.00 |
| Rank 100 | 1st Serve Return Points Won | 0.22 | 9 | 0.02 | 8.92 | < .001 | .214 | 80.24 | 1.00 |
| | 2nd Serve Return Points Won | 0.01 | 9 | 0.01 | 3.03 | .002 | .085 | 27.27 | .97 |
| | Break Points Converted | 1.13 | 9 | 0.13 | 4.65 | < .001 | .125 | 41.82 | 1.00 |
| | Return Games Won | 0.50 | 9 | 0.06 | 10.16 | < .001 | .237 | 91.41 | 1.00 |
| | Return Points Won | 0.15 | 9 | 0.02 | 8.51 | < .001 | .207 | 76.61 | 1.00 |
| | Total Points Won | 0.15 | 9 | 0.02 | 20.92 | < .001 | .390 | 188.24 | 1.00 |
| Rank 300 | 1st Serve Return Points Won | 0.06 | 3 | 0.02 | 6.67 | < .001 | .063 | 20.02 | .97 |
| | 2nd Serve Return Points Won | 0.06 | 3 | 0.02 | 4.39 | .005 | .042 | 13.18 | .87 |
| | Break Points Converted | 0.20 | 3 | 0.07 | 2.25 | .083 | .022 | 6.74 | .57 |
| | Return Games Won | 0.02 | 3 | 0.07 | 10.43 | < .001 | .094 | 31.29 | 1.00 |
| | Return Points Won | 0.05 | 3 | 0.02 | 7.03 | < .001 | .066 | 21.09 | .98 |
| | Total Points Won | 0.09 | 3 | 0.03 | 28.36 | < .001 | .221 | 85.07 | 1.00 |
| Rank 500 | 1st Serve Return Points Won | 0.02 | 2 | 0.01 | 2.52 | .082 | .016 | 5.04 | .50 |
| | 2nd Serve Return Points Won | 0.03 | 2 | 0.01 | 2.94 | .054 | .019 | 5.88 | .57 |
| | Break Points Converted | 0.08 | 2 | 0.04 | 1.34 | .263 | .009 | 2.68 | .29 |
| | Return Games Won | 0.05 | 2 | 0.03 | 3.92 | .021 | .025 | 7.83 | .70 |
| | Return Points Won | 0.01 | 2 | 0.01 | 2.16 | .117 | .014 | 4.32 | .44 |
| | Total Points Won | 0.04 | 2 | 0.00 | 17.42 | < .001 | .104 | 34.85 | 1.00 |

NCP, Non-Centrality Parameter; 1-β, Observed Power.

these variables are considerably more critical. It also suggests that in today's professional league, while anthropometric characteristics are important, gaming-related characteristics are even more significant.

As the effects of all the research-dependent variables were confirmed across all IVs, our interest was piqued regarding which individual DV influences the various levels of ATP status. An overview of the results of these univariate tests can be found in Table 3.

Table 3 reveals that the majority of univariate test results for different MANOVAs remained statistically significant even after applying the Bonferroni correction. Consequently, it is plausible to assert that individual DV were influenced by the IVs within the context of MANOVA. For a more intricate examination of the specific effects of each IV on each DV individually, rather than considering them as a collective group, as performed in the MANOVA, we can denote effects ranging from trivial (.009) to substantial (.390) in terms of the IV's contribution to the variance in the DV. Eta-squared ($\eta^2_p$) aids in gauging the magnitude of this influence. Therefore, we can affirm that we have identified precisely 14 large, 12 medium, and 17 small effects. Among the 14 variables that exhibited large effects, none were among the anthropometric DVs, five pertained to return percentages (Return % DV), and nine were associated with service-related variables (Service DV). To illustrate it, variables such as Weight (for Rank 100) and Break Points Converted (for Rank 500) explained very little variance in the DVs (0.9%), whereas Total Points Won explained 39.0% of the variance in the DVs. After obtaining these positive and statistically significant findings, it is common to conduct a post hoc test to further explore the statistically significant observations. Based on the Scheffe post hoc test, the application of the Bonferroni correction, and an assessment of descriptive statistics, the following conclusions can be drawn.

In case of Anthropometric-base variables, it was found that tennis players ranked 1–100 (26.86±4.22 years) are older than those ranked above 900 (22.56±4.12 years), and players ranked 1–300 (26.21±4.37 years) are older than players in the range of 601–900 (24.03±4.43 years) and above 900 (22.56±4.12 years). Furthermore, players ranked 1–100 (25.87±4.53 years) are older than players who achieved rankings of 501–1000 (23.95±4.27 years) and above 1000 in the ATP rankings (22.47±4.11 years). This finding can be interpreted as an indication that with increasing age, the probability of achieving a better ranking also increases. Tennis players ranked 1–500 (78.95±6.64 kg) had a higher weight than players ranked above 1000 (77.39±7.02 kg). This was the only statistically significant result in the case of the Weight variable. Players ranked 1–100 (187.77±7.12) were taller than players ranked above 900 (183.43 ±6.74); likewise, players ranked 1–300 (185.39±6.76) were taller than players ranked above 900 (183.43±6.74). Since players ranked 1–500 (185.25±6.81) were taller than players ranked 501– 1000 (183.58±6.89) and above 1000 (183.25±6.72), it can be inferred that player height may be a suitable factor for predicting their ranking. The four variables with the strongest effects were selected and displayed in Fig 1.

After Anthropometric-base variables, Service-based variables were analysed. Interestingly, the variable 1st serve (%) did not yield statistically significant results, possibly due to the fact that most players focus on this variable during their training, and as a result, there are no significant differences among the various categorical variations of the ATP ranking. In the case of 1st service points won (%), players ranked 1–100 (0.72±0.05) achieved better results than players ranked above 1000 (0.54±0.09). This trend continues even for players ranked 1–300 (0.70 ±0.06), who achieved better results than players ranked above 900 (0.54±0.09). Therefore, this appears to be a variable with a significantly predictive value. Break Points Saved is one of the variables where differences were not visible between the first and last research categories. Players ranked 1–100 (0.51±0.03) had better results than players ranked 201–300 (0.45±0.07) and 401–500 (0.42±0.09). However, it is important to note that this lack of difference is primarily

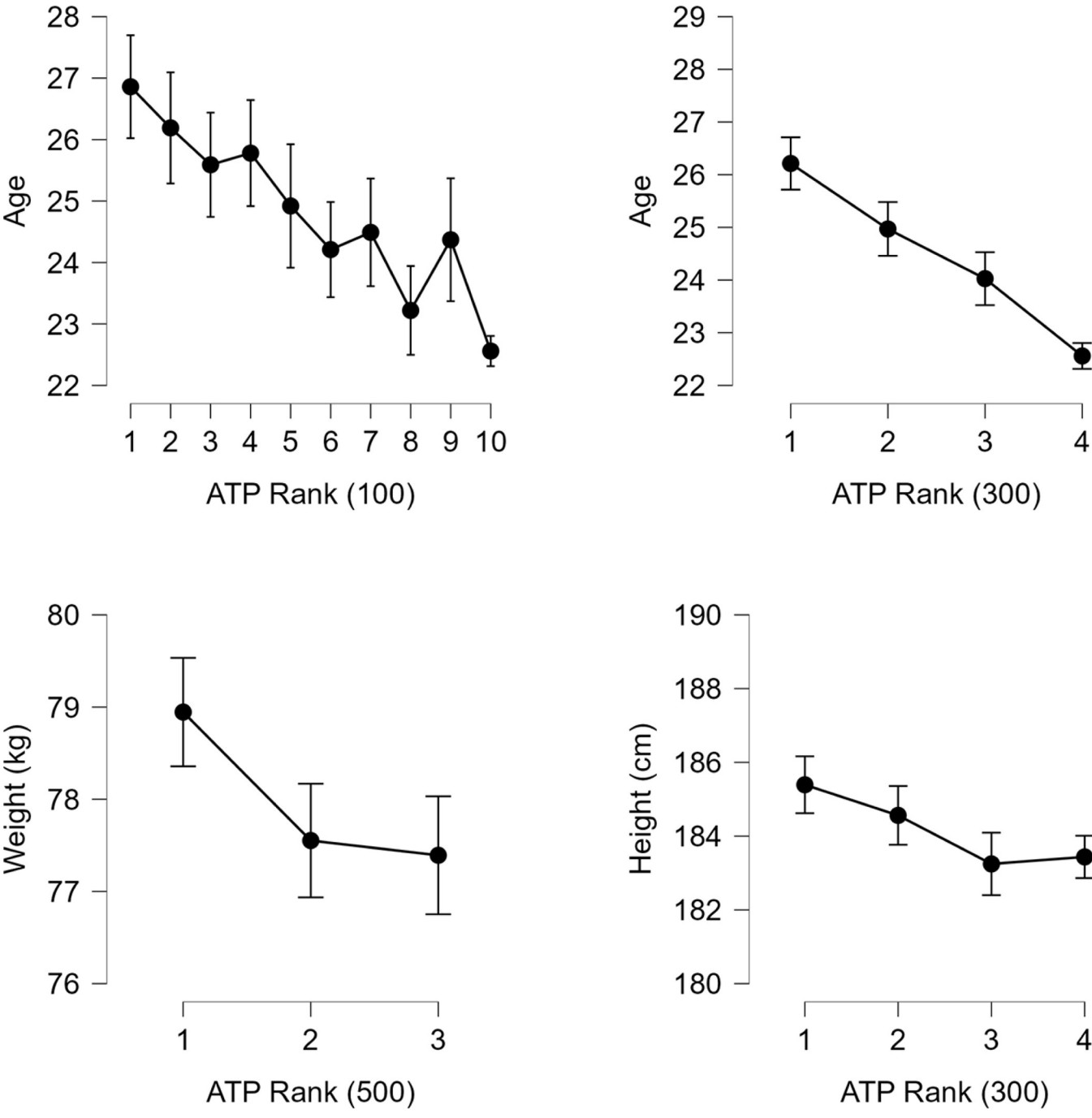

**Fig 1. Comparison of selected ATP rank and anthropometric-based characteristics.**

due to statistical significance, as the worst values were achieved by tennis players ranked above 900 (0.40±0.04), but this is based on only five valid data points. A similar pattern was observed in the case of differences between players ranked 1–300 (0.49±0.06) and 301–600 (0.45±0.08), where players ranked above 900 (0.40±0.04) were not included in the calculations due to the low number of data points. Regarding Service Games Played, a significant separation can be observed among players' ranks. Players ranked above 900 (0.32±0.32) had inferior results

compared to players ranked 1–100 (0.62±0.4), 101–200 (0.58±0.10), 201–300 (0.57±0.13), 301–400 (0.49±0.18), 401–500 (0.46±0.22), 501–600 (0.56±0.16), and 701–800 (0.37±0.24). A similar trend can also be observed when dividing the data according to ATP (300) and ATP (500), where the last categories achieved the lowest results. The variable Service games won provided interesting insights into player statistics and ATP ranking. Players ranked 1–100 (0.80±0.06) exhibited better results than players ranked 301–400 (0.67±0.016), 401–500 (0.64 ±0.17), and those ranked above 900 (0.46±0.16). Additionally, players ranked 101–200 (0.74 ±0.10) achieved superior results compared to players ranked above 900 (0.46±0.16). Therefore, it was not surprising to observe a noticeable downward trend when dividing the data by ATP (300). More precisely, players ranked 1–300 (0.76±0.09) demonstrated better results than players ranked 301–600 (0.66±0.15), 601–900 (0.59±0.22), and even those ranked above 900 (0.46 ±0.16). A similar trend emerged when dividing the data by ATP (500), with players ranked 1–500 (0.74±0.11) achieving superior results than players ranked 501–1000 (0.63±0.17) and those ranked above 1000 (0.45±0.18). Consequently, this variable also exhibits great potential for prediction. The variable Total service points won exhibited the trend described above, where players ranked 1–100 (0.64±0.05) achieved better results than players ranked above 900 (0.49±0.06). Further analysis revealed that players ranked 1–100 (0.64±0.05) achieved superior results than players ranked 101–200 (0.58±0.07) and those ranked above 900 (0.49±0.06). While tennis players ranked 201–300 exhibited a similar trend (i.e., worse results than the top players), due to the high degree of uncertainty (given the significant variability in the data), this effect was not statistically significant. Nonetheless, players ranked 1–500 (0.61±0.05) achieved better results than players ranked above 1000 (0.48±0.07). The four variables with the strongest effects were selected and displayed in Fig 2.

In the case of variables falling under the category of Return %, fewer statistically significant variables were identified compared to the previously studied areas (Anthropometric and Service %). As one of the two main variables (along with Total Points Won), 1st Serve Return Points Won had the largest differences not between top and bottom ranks, but between ranks 1–100 (0.29±0.03) and 201–300 (0.23±0.07), as well as between players from 101–200 (0.28±0.04) and 301–400 (0.24±0.06). However, it is important to note that this variable, despite being identified as one of the main variables, has low predictive power, as its mean values exhibit oscillating tendencies. Other statistically significant values were observed in the variables Return Games Won and Return Points Won rank, where in both cases, players ranked 1–100 (0.21±0.05 and 0.37±0.03, respectively) outperformed those ranked above 900 (0.06±0.13 and 0.31±0.06). Another main variable in this section with greater predictive capability than the previously mentioned variable was Total Points Won, where players from 1–100 (0.50±0.02) achieved better results than tennis players from 101– 200 (0.48±0.03), 201–300 (0.46±0.04), 301–400 (0.46±0.04), 401–500 (0.45±0.04), and those ranked above 900 (0.40±0.06). Notably, players from 101–200 (0.48±0.03) achieved better values than those ranked above 900 (0.40±0.06). This trend was also evident between ranks 1–300 (0.49±0.03) and 301–600 (0.46±0.04), 601–900 (0.43±0.05), and those ranked above 900 (0.40±0.06). In this context, this variable stands out as having the best predictive capability, as the mean values for this variable (Total Points Won) exhibit a linear decreasing tendency, after deleting outliers from the dataset. It is worth noting that even among the top 5 players (e.g., Novak Djokovic), there was an outlier value, indicating significantly better performance compared to other players (0.55). However, this outlier value was not evident in the last division, where players ranked 1–500 (0.48±0.03) achieved better values than players ranked above 1000 (0.40±0.07). The four variables with the strongest effects were selected and displayed in Fig 3.

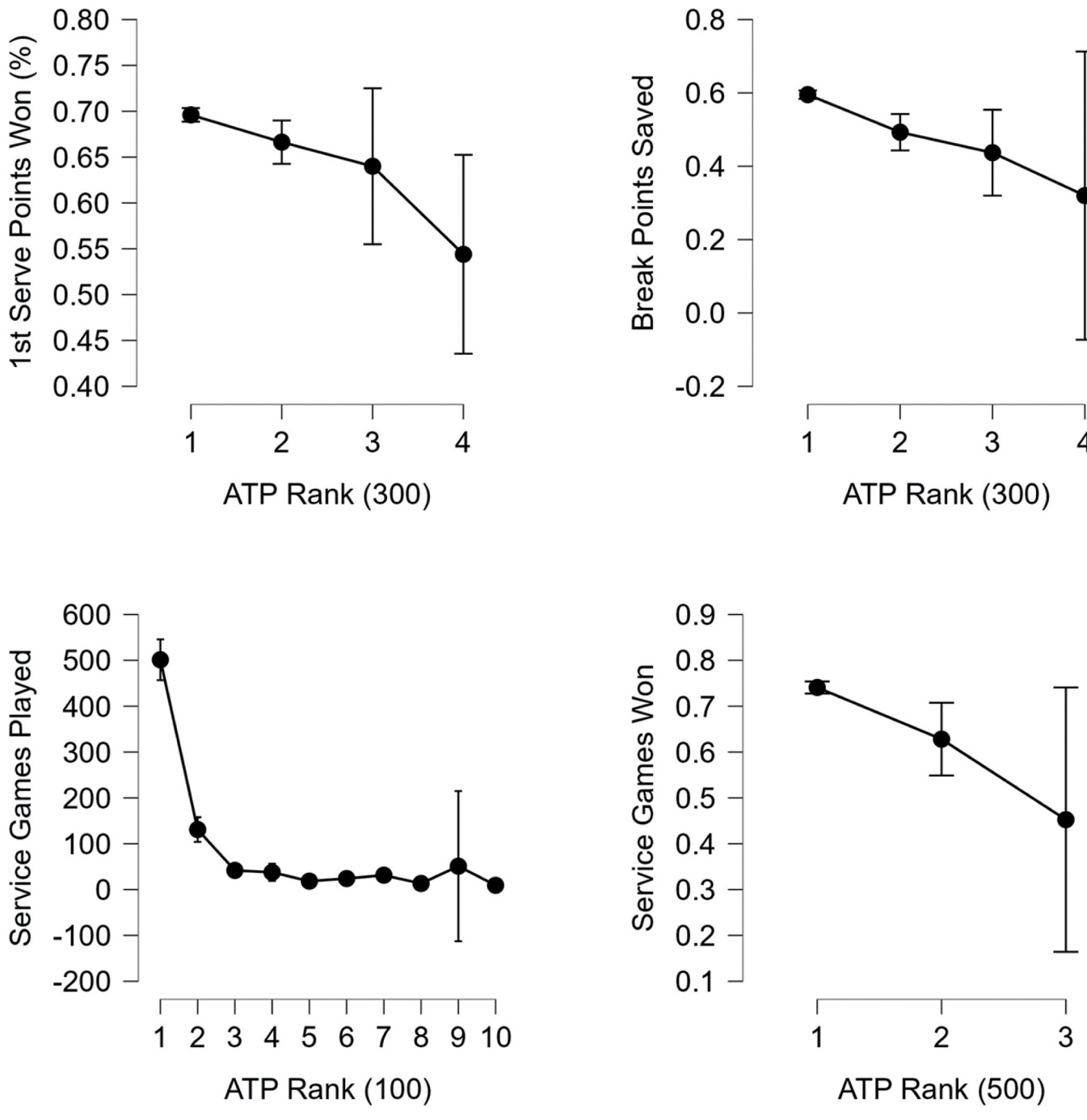

**Fig 2. Comparison of selected ATP rank and service-base characteristics.**

### Multinomial Logistic Regression results for ATP rank outcomes

To evaluate the overall contribution of each IV to the model, Likelihood Ratio tests were used in which variables Age, Height in ATP 10, ATP (300) and ATP (500) were statistically significant predictors in the model focusing on anthropometric characteristics. In case of Service, we found statistically significant results in variables 1st Serve Points Won, 2nd Serve Points Won at ATP (100); 1st Serve Points Won, 2nd Serve Points Won, Service Games Played at ATP

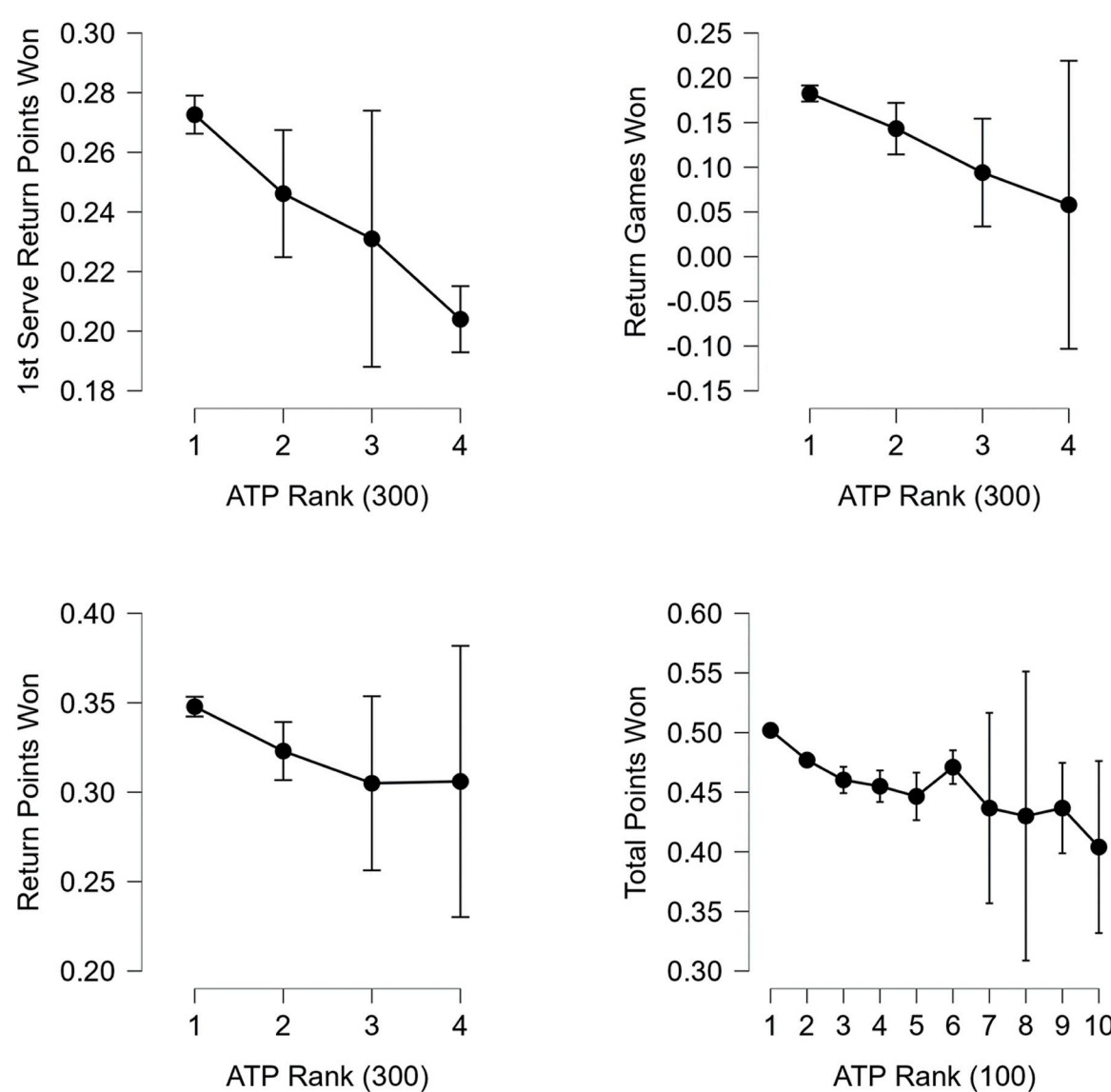

**Fig 3. Comparison of selected ATP rank and return-base characteristics.**

(300); 1st Serve Points Won, Service Games Played at ATP (500). In case of Return we found only one statistically significant result, more precisely in variable Total Points Won for data categorization according to ATP (100). The Bonferroni correction was applied to all reported results.

The final results of MLR, based on Parameter Estimates (After Bonferroni correction), reveal five statistically significant variables, more precisely Age at ATP 100, 300 and 500 categorizations; Height at ATP (100) and Total Points Won at ATP (100), relatively to the (last) reference category. Scheffe's Post hoc test, after the Bonferroni correction, reveals statistically significant results for players Age at 1.-100. rank (OR = 1.16; 95% CI = 1.10,1.21), 101.-200. rank (OR = 1.13; 95% CI = 1.07, 1.18), 201.-300. rank (OR = 1.09; 95% CI = 1.04, 1.14), 301.-400. rank (OR = 1.11, 95% CI = 1.06, 1.16), 1.-300. rank (OR = 1.12, 95% CI = 1.09, 1.16), 301.-600. rank (OR = 1.06; 95%CI = 1.03, 1.10), 1.-500. rank (OR = 1.10; 95% CI = 1.07, 1.14); Height at 1.-100. rank (OR = 1.11, 95% CI = 1.06, 1.16). In the case of Service, no statistically

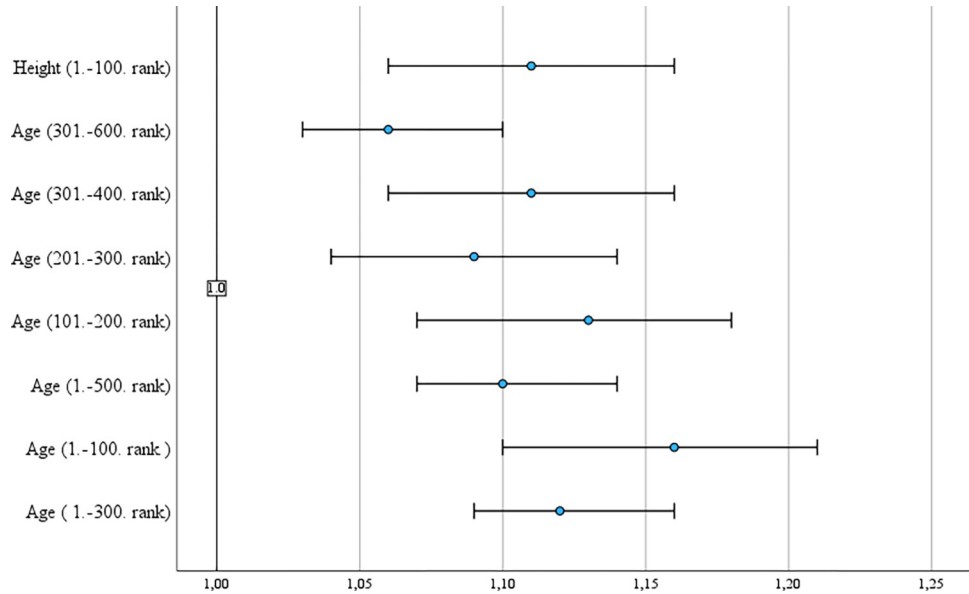

**Fig 4. Comprehensive overview of statistically significant results from Multinomial Logistic Regression (MLR) including Odds Ratios (OR) and 95% Confidence Intervals (CI), with reference line for none effect.**

significant result was found, but one statistically significant result was documented in Return players statistics, more precisely, Total Points Won at 1.-100. rank (OR = 8.53E+41; 95% CI = 4.74029E+18, 1.53E+65). Because every statistically significant result contains positive beta coefficient results. For example, in the case of the variable Age, this indicates that older ATP tennis players are more likely to be in the 1–100 rank. The Odds ratio for this variable was 1.157, indicating that for each unit increase in age, the odds of being ranked 1. to 100. change by a factor of 1.16 (95% CI = 1.10–1.21). In other words, older ATP tennis players in the 2022 season had more chance to be among top 100 ATP players at the end of season. Similar findings (with different intensity) can also be demonstrated in the case of 101.-200 rank, 201.-300. rank, 301.-400. rank, 1.-300. rank, 301.-600. rank, 1.-500. rank, as well as in the case of height, where for each unit increase in height, the probability of being among the top 100 players changes by a factor of 1.11 (95% CI = 1.06–1.16). In other words, with each 1cm, the chance increases by 11.0%. For better visualization, the above-mentioned effects are presented in Fig 4 using Odds Ratios and a 95% Confidence Interval. The graph also includes a null effect line.

## Neural Network analysis of multivariate outcomes associated with ATP rank

Multiple models were generated and subsequently analysed, yet only the most precise Neural Network (NN) models are depicted in Figs 5 and 6 (by results of test validation, y-axis), along with their corresponding details in Table 4. It is essential to highlight that unlike the other examined response variables, Prize Money and Points were not categorical; instead, they are continuous variables. Consequently, a lower level of validity may be anticipated for these particular variables.

The average accuracy across all 35 generated models stood at 65.80±21.4. In the detailed analysis of the best generated models categorized by predictors (Fig 5), it is evident that results in the poorest models ranged from 24.2% (Return %) to 51.5% (Service, Return), while in the

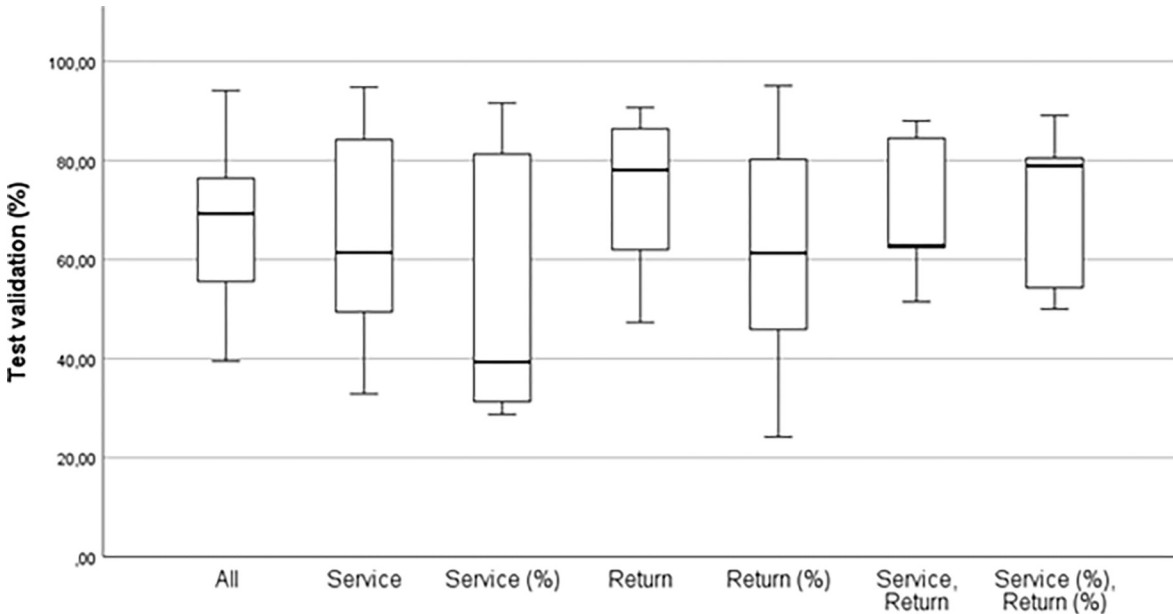

**Fig 5. An overview of Neural Network test validation according predictors.**

best models, they ranged from 88.0% (Service, Return) to 94.8% (Service). However, the predictor Return attained the best average result (72.9±18.0). Due to the inconsistency and insufficient validity of the results, we must state that we found ambiguous outcomes in the predictor-based categorization. The same, cannot be said based on the detailed analysis of the test validity results categorized by response (Fig 6). In this case, we found that the least valid models ranged

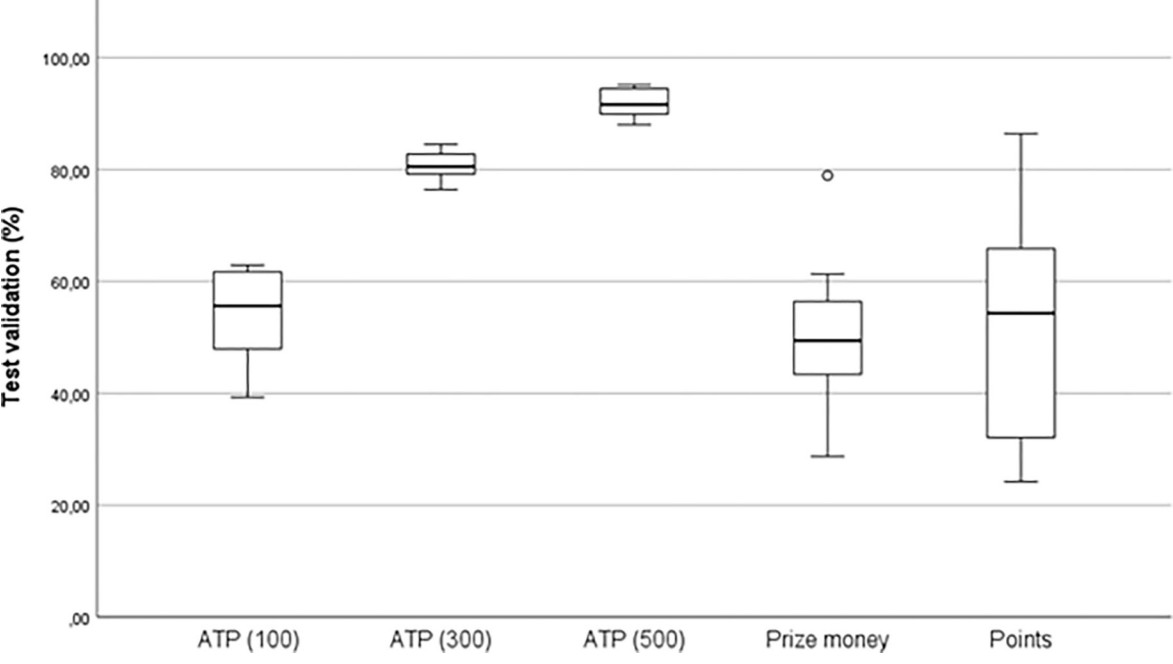

**Fig 6. An overview of Neural Network test validation according response.**

from 24.2% (Points) to 88.0% (ATP 500). Models with the highest accuracy ranged from 62.9% (ATP 100) to 95.1% (ATP 500), and the model with the highest average value was ATP 500 (91.9±2.8). From this description, the clear dominance of predictive model accuracy in ATP 500 is apparent, where even the worst generated model achieved better accuracy than the best model in other generated AI models (diff = 1.6%).

When employing Service (%) and Return (%) as predictors to estimate Prize Money, this model achieved a test validity of 78.9%, albeit being an outlier value. With a larger dataset or more meticulous fine-tuning, this model could potentially attain higher accuracy. Consequently, it cannot be categorically deemed unusable, like the points predicting the model. Which, due to its considerable variability in test validity, we advise against its practical application.

When arranging all models based on their test validity, the models pertaining to ATP Rank (500) occupy the top (seven) positions, according to the test validation. These models are, therefore, unequivocally the most reliable and, in practice, the most useful models (mean = 91.91; SD = 2.83; min = 88.00; max = 95.1; $\gamma_1$ = -0.18; $\gamma_2$ = -1.80). The models associated with ATP (300) rank follow in the second position (mean = 80.74; SD = 2.96; min = 76.40; max = 84.50; $\gamma_1$ = -0.05; $\gamma_2$ = -0.86). However, it is worth noting that even the poorest-performing model within ATP (500) demonstrated better accuracy than the best-performing ATP (300) model (a difference of 1.6%), as mentioned earlier.

The AUC serves as a highly suitable metric for monitoring and interpreting results with categorical outcomes, as it quantifies predictive strength across different categories. For instance, using Model 1.4 as an example, it can be observed that the model can reliably predict categories 1 to 100 Rank based on the Return variable (AUC = .972). Conversely, for the outcome in the 801 to 900 Rank category, the prediction is unreliable (AUC = .528) and can be likened to a coin toss with a 50:50 chance of accuracy. Thus, it is important to emphasize that even though the best model achieved commendable validity (Model 3.5 with 95.1% accuracy), the AUC value could be interpreted as ranging from poor to fair. The second-most accurate model also faces a similar challenge (Model 3.2, failing to achieve excellence in AUC). It is only the third-most valid model, Model 3.1, with 94.1% accuracy, that can be classified as achieving excellent AUC across all outcome categories. The NN Model 3.1 demonstrated superior performance and was subsequently selected for fine-tuning, resulting in a training phase accuracy of 95.2% and a testing phase accuracy of 96.8%, with an AUC ranging from 0.933 to 0.993. Displayed in Fig 7 in ROC Curve form. The post-fine-tuning outcome values are presented in Table 4. The scheme of the neural network for model 3.1 after fine-tuning, illustrating the number and degree of relationship of the input variables, the number of hidden layers, types of activation functions, and the output variable, is included in the S1 Fig.

The best NN model predicting continuous responses was Model 5.4, which predicted Points using all Return variables with an accuracy of 86.4%. Since the second least accurate model achieved only 78.9% accuracy, it can be concluded that Model 5.4 is unequivocally the best model for predicting continuous data.

## Discussion

This study objectives involved employing classical and advanced statistical approach to quantify the importance of anthropometric, service-based, and return-based variables in relation to the categorized ATP rank of tennis players registered at the ATP association in season 2023.

Based on results from MANOVA testing, a strong effect size was found in the case of two DV. More precisely, 17.8% of the variance in the DV Service (%) can explained by the IV ATP (500) and 15.3% of the variance in the Service (%) can explained by the IV variable ATP (300).

**Table 4. An overview of the most accurate Neural Network models.**

| Model | Response | Predictor | Validation (%) | | AUC | Independent Var. Importance | |
|---|---|---|---|---|---|---|---|
| | | | Training | Test | | Lowest (%) | Highest (%) |
| 1.1 | ATP (100) | All | 64.3 | 55.6 | .818-.977 | Break Points Converted (.026) | Age (.078) |
| 1.2 | | Service | 60.3 | 61.4 | .610-.999 | 2nd Serve Points Won (.075) | Service Games Played (.139) |
| 1.3 | | Service (%) | 44.2 | 39.3 | .333-.999 | Break Points Saved (.121) | Service Games Won (.246) |
| 1.4 | | Return | 59.8 | 62.0 | .528-.972 | Break Points Converted (.072) | Return Points Won (.198) |
| 1.5 | | Return (%) | 48.0 | 45.9 | .677-.983 | Break Points Converted (.089) | Total Points Won (.289) |
| 1.6 | | Service, Return | 54.5 | 62.9 | .437-.970 | Break Points Faced (.090) | Break Points Opportunities (.288) |
| 1.7 | | Service, Return (%) | 53.4 | 50.0 | .609-.999 | 1st Serve Return Points Won (.053) | Return Games Won (.155) |
| 2.1 | ATP (300) | All | 85.4 | 76.4 | .891-.944 | 2nd Serve Points Won (.021) | Tournament Played (.129) |
| 2.2 | | Service | 81.7 | 84.2 | .578-.999 | Double Faults (.065) | Service Games Won (.207) |
| 2.3 | | Service (%) | 76.8 | 81.3 | .689-.999 | 2nd Serve Points Won (.089) | Service Games Won (.294) |
| 2.4 | | Return | 82.1 | 78.1 | .596-.960 | Return Games Won (.053) | Return Points Won (.198) |
| 2.5 | | Return (%) | 77.9 | 80.2 | .786-.997 | Return Games Won (.092) | Total Points Won (.287) |
| 2.6 | | Service, Return | 77.0 | 84.5 | .803-.954 | Aces (.097) | Break Points Opportunities (.245) |
| 2.7 | | Service, Return (%) | 80.9 | 80.5 | .486-.888 | Break Points Converted (.048) | Break Points Saved (.196) |
| 3.1[a] | ATP (500) | All | 95.2 | 96.8 | .933-.993 | Break Points Opportunities (.018) | Tournament Played (.176) |
| 3.2 | | Service | 90.4 | 94.8 | .166-.999 | Break Points Faced (.058) | Break Points Saved (.179) |
| 3.3 | | Service (%) | 91.9 | 91.6 | .808-.999 | Break Points Saved (.101) | Service Games Won (.321) |
| 3.4 | | Return | 92.7 | 90.7 | .833-.892 | Return Games Won (.061) | Return Points Won (.169) |
| 3.5 | | Return (%) | 91.0 | 95.1 | .746-.682 | Return Points Won (.113) | Return Games Won (.231) |
| 3.6 | | Service, Return | 93.2 | 88.0 | .770-.926 | Double Faults (.096) | Break Points Opportunities (.234) |
| 3.7 | | Service, Return (%) | 92.9 | 89.1 | .786-.999 | 2nd Serve Points Won (.035) | Total Service Points Won (.161) |
| 4.1 | Prize money | All | 86.8 | 39.5 | - | 1st Serve Points Won (.022) | Break Points Opportunities (.212) |
| 4.2 | | Service | 62.6 | 49.4 | - | Service Games Played (.033) | Break Points Faced (.142) |
| 4.3 | | Service (%) | 26.5 | 28.7 | - | 1st Serve Points Won (.114) | 2nd Serve Points Won (.283) |
| 4.4 | | Return | 47.3 | 47.3 | - | Return Games Won (.028) | Total Points Won (.341) |
| 4.5 | | Return (%) | 47.8 | 61.3 | - | 2nd Serve Return Points Won (.033) | Total Points Won (.625) |
| 4.6 | | Service, Return | 91.9 | 51.5 | - | Double Faults (.051) | Break Points Opportunities (.546) |
| 4.7 | | Service, Return (%) | 72.0 | 78.9 | - | Total Points Won (.033) | Break Points Saved (.284) |
| 5.1 | Points | All | 92.1 | 69.3 | | Service Games Played (.023) | Break Points Opportunities (.208) |
| 5.2 | | Service | 55.2 | 32.9 | - | Service Games Won (.023) | Aces (.228) |
| 5.3 | | Service (%) | 21.5 | 31.3 | - | Break Points Saved (.086) | 1st Serve (.282) |
| 5.4 | | Return | 86.4 | 86.4 | - | 2nd Serve Return Points Won (.038) | Break Points Opportunities (.328) |
| 5.5 | | Return (%) | 41.0 | 24.2 | - | Break Points Converted (.060) | Total Points Won (.413) |
| 5.6 | | Service, Return | 88.2 | 62.4 | - | Double Faults (.029) | Break Points Opportunities (.567) |
| 5.7 | | Service, Return (%) | 67.5 | 54.3 | - | 1st Serve Points Won (.033) | 1st Serve Return Points Won (.176) |

Response variables Prize money and Points were treated as continuous variables (AUC results were not included); AUC is present in formed of range (minimum to maximum).

[a]Results after post-fine-tuning.

Service (in %) explained the largest variation in comparison with the other two DV (anthropometric characteristics and returns (in %).

A post hoc test for MANOVA reveals multiple statistically significant findings among categorized ATP ranks in the case of Age, Weight, and hand eight, where top players were older, heavier, and taller than players from other categorized ATP ranks (mostly the bottom ranks). For example, Age specificity in trainability should be considered when designing programs for long-term athlete development [8]. Analysing anthropometric variables is crucial in

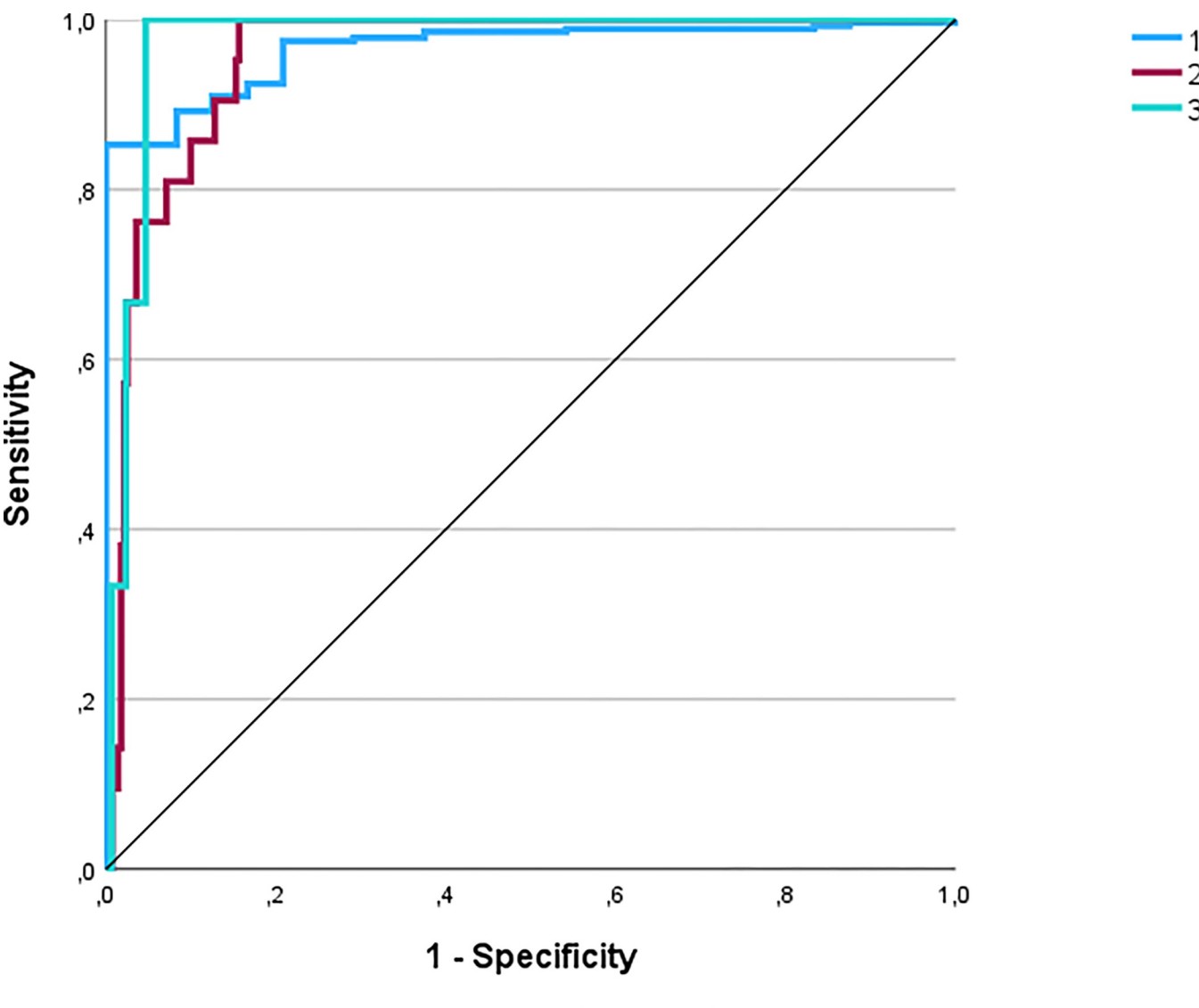

**Fig 7. ROC Curve for best performing Model 3.1 (After post-fine-tuning).**

performance analysis because a player's height has a significant impact on serve and return outcome [50], and also serve speed [23], which is a pivotal variable in predicting match success [31], or can be used as a predictor indicating a change in the opponent's game strategies [33]. In addition, these variables also affect players' fatigue and thus players' tactics [37]. In this study, we also found that the average height of all registered ATP players in the 2022 season was 184.06±6.86 cm (max = 211 cm), with slightly negative Skewness of -0.19, indicating that players tend to have higher than average height. From a total of 1389 valid values, 473 (34.1%) players were taller than 186 cm, and 296 (21.3%) had less than 180 cm, which can negatively affect their chance to win a match [32], even if taller players had higher speed on 1st and 2nd serve [50]. Therefore, only 620 (44.6%) ATP players had the ideal height. It is important to note that there are also limits of effectiveness in the case of anthropometric characteristics and that these reference values are based on data from 1991–2008 (Grand Slam, men's singles).

In the case of the Service-base variable, we did not find statistically significant results for 1st service (%), which contradicts the conclusions of other studies [31, 34, 35]. This can be explained by the general awareness of the importance of this variable, which tennis players become aware of and approach their game strategy and training accordingly. In contrast, we found that top tennis players have better results than others (mainly than the bottom ranked players) in the variables 1st service points won (%), Break Points Saved, Service games won, and Total service points won. These conclusions were already reported by other studies [31, 32, 38, 45, 50] and although they did not always analyze the same type of variables, we can characterize them as their alternatives. The dynamics of service variables can also be illustrated by the statistics at Wimbledon in the years 2002–2015, where service game efficiency (service games won, 1st and 2nd serve points won, and ace (%)) increased and double fault (%) decreased. At the same time, serve success (service game, 1st and 2nd serve points winning (%)) and serve performance (ace and double fault (%)) appear to be more important variables in the 2nd week of the tournament than in the 1st [51]. It is important to note that this study only dealt with service-base variables (not return-base).

In the case of the return-based variable, we found fewer statistically significant results compared to the Anthropometric-based and Service-based variables, which contradicts the study by Cui et al. [37], who observed a decreasing trend in the service-based variable and an increase in return-based variables during set-to-set variations in performance. These conclusions indicate the importance of return-based variables within one match, which, however, are less important within the analysis focused on another time period (performance). As key factors to determine the best players, we can mention the variables 1st Serve Return Points Won, Return Games Won, Return Points Won rank, Total Points Won, because in all these variables, the top players achieved better results than other tennis players from the ATP ranking. These conclusions are in agreement with other studies dealing with a similar topic [27, 34, 36, 46, 49]. It is also possible to conclude that during the match, there is a decrease in the importance of service-based (net play, running) variables and an increase in the importance of return-based variables in the last sets compared to the initial sets. This can be attributed to fatigue, the choice of tactics, and pacing [37], showing the dynamics of this variable.

However, anthropometric characteristics did not prove to be sufficiently good predictors, in the case of MLR, because we found only three statistically significant variables (in total of five different levels). Age, height, and Total Points Won are suitable variables for predicting ATP ranking. These results (compared to the results of MANOVA) can indicate that MLR is more sensitive to specific data characteristics, more complex patterns, or a strong correlation between predictors [58, 59]. Therefore, it is not appropriate to reduce the importance and practical suitability of statistically significant results from the MANOVA test on the basis of statistically insignificant results from the MLR. These are different methods by which test different things. Because inferential statistical technique such as MLR has shown problems with robustness within high-dimensional models, Machine learning techniques such as NN are more suitable for those tasks [60, 61].

Advancements in technology have enabled in-depth technical analyses. Makino et al. [40] utilized machine learning techniques to predict point winners, emphasizing the significance of specific shot analysis. Mergheş et al. [49] discussed the intricacies of technical skills, particularly focusing on the return of serve as a pivotal shot. Gillet et al. [27] emphasized the enduring importance of serves and serve-returns, even on slower surfaces like clay, in influencing match outcomes. The conclusions from this study also indicate that service-based and return-based variables are important and suitable variables for predicting outcomes. However, a much more important factor is the type and number of output variables for which the model was trained and tested, which is also pointed out by other authors [62–64]. Future NN architects should,

therefore, consider the suitability and adequacy of the output variable, for which we recommend setting the smallest possible number of categorical output variables, e.g. Win/Loss; strategies initial/intermediate/intensive part of the match; ATP Rank (500). Then they can try to gradually increase the number of categories, or switch to continuous data prediction. If we compare our best-performing AI model 3.1 (predicting ATP (500) with all researched variables), which achieved 96.8% accuracy with an AUC range of 0.933 to 0.993 (after post-fine-tuning), we can claim that our AI model, utilizing Neural Networks, achieved superior predictive accuracy compared to Makino et al.'s [40] model. Makino et al. [40] employed L1-regularized logistic regression and achieved an accuracy of 66.5% with an AUC of 0.689 for predicting winners and valuable features. However, it is worth noting that our model has limited applicability early in the season, becoming more valid as the season progresses.

The popularity of tennis from the perspective of players, coaches, and stakeholders is influenced by the prospect of financial rewards in the form of prize money, which increases alongside competitiveness [65]. However, we found that 87.73% of the ATP players will not reach the average Prize money. Similar conclusions are also reported by Balliauw et al. [66], when they state that professional tennis players who are below the level of the top 250 rank have difficulty covering the financial expenses associated with their profession.

## Limitations and directions for future research

Although the current study was based on 28 variables and encompassed a substantial amount of data (20,040 data points), there are numerous other approaches and variables that were not investigated in this study, such as the average serving speed or time-oriented variables. One future research direction could be the development of a predictive model that works with the game characteristics of tournament or match winners. However, this type of AI model would require training and testing using multiple variables that were not necessary to track in this study, such as the type of court, because this has a profound impact on players' strategies. Tudor et al. [33] demonstrated varying player approaches on fast and slow courts, emphasizing adaptability. Cui et al. [36] discussed court-specific tactics, underlining the importance of baseline play and relative quality in diverse court environments. Furthermore, we recommend that variables such as tournament rounds (week), and game statistics from the initial, intermediate, and intensive part of the match should also be included in this AI model because these factors affect performance, fatigue, pacing, and game strategy and thus on the outcome of the match [37, 38]. This AI model could thus assist in determining an appropriate (initial, backup, or alternative) game strategy based on these pieces of information.

As a significant variable that could profoundly influence the accuracy and error rate of future AI models, we consider the technique of disguising strokes, a common strategy in elite-level tennis tournaments [3]. For this highly specific AI model, it would be appropriate to develop our own specialized CNN, capable of processing photographs and natural language, as well as analyzing, categorizing, and quantifying data acquired from video footage. The results from this model could subsequently serve as a control variable for another AI model and thus reduce its error rate and increase model accuracy. A similar type of AI model could also be used to predict first and second serve related to the target zone for ball impact [67].

In the current study, we focused solely on professional male tennis players and did not include women or juniors due to significant disparities in performance between professional and junior players, as emphasized by Kovalchik and Reid [35]. The research conducted by Reid et al. [46] and Kovalchik and Reid [35] showed that professional players have a significant advantage in serving, with male professionals winning 4% more points and female professionals winning 2% more points on serve compared to their junior counterparts. These differences

in stroke dynamics, particularly in relation to the first serve and serve-return, and movement speeds, were highlighted by the study of Cui et al. [36], who stress the need for gender-specific training, practice designs, and AI modeling. Stare et al. [45] demonstrated that professional players focus more on offensive baseline play and achieve a higher number of aces with their first serve. Pereira et al. [47] also noted that young players spend more time in the serving phase, whereas professionals spend more time in the returning phase. These differences in gameplay and physical characteristics of shots between junior and professional tennis players underscore the necessity for differentiated training methods, gender-specific, and performance-level specific approaches, which is the primary reason for limiting this study to professional male tennis players. Otherwise, one would expect a much larger error rate in inferential statistics and a lower predictive accuracy of AI models.

## Conclusion

In conclusion, our study has yielded valuable insights into predicting tennis players' rankings on the ATP Rank through various analytical approaches.

From the results of descriptive statistics, it can be concluded that there are significant implications for tennis players. The data suggests that focusing on accumulating minimal points across multiple tournaments, without a comprehensive approach, may offer valuable experience to players. However, this strategy does not lead to substantial financial gains or higher rankings. Players aspiring to achieve higher standings in the ATP rankings should consider adopting a more strategic and focused approach to their tournament participation and gameplay. While it might seem logical to assume that the top players earn the most Prize money, the data provides evidence of a significant disparity in Prize money across the dataset. This is indicated by the Skewness and Kurtosis coefficients, reflecting a right-skewed data distribution and a more peaked frequency distribution, respectively. These results indicate that tennis can be a highly profitable sport, but only for a small percentage of the best players (%). For this select group, tournament income is sufficient. However, for the remaining players, it proved to be insufficient in the year 2022.

In the MANOVA analysis, we observed numerous statistically significant differences among various player statistics, emphasizing the need to consider multiple variables rather than focusing on a single one. This multifaceted approach allows us to categorize variables based on their importance, providing a more comprehensive understanding of their impact on ATP rankings. Notably, certain performance metrics, such as 1st Serve Return Points Won, exhibited limited predictive power due to their fluctuating nature. In contrast, other metrics, including 1st Serve Return Points Won, Return Games Won, and Return Points Won rank, consistently distinguished players in higher rankings from those in lower positions. Total Points Won emerged as a robust predictor, demonstrating a clear linear trend towards superior performance in lower-ranked players.

The MLR analysis examined the determinants of ATP Rank outcomes, focusing on anthropometric characteristics, service-related factors, and return statistics. Notably, Age and Height variables displayed statistical significance within different ATP Rank categorizations, highlighting their substantial influence on player rankings. In the realm of service-related factors, specific metrics such as 1st Serve Points Won, 2nd Serve Points Won, and Service Games Played exhibited statistical significance in various ATP categorizations (100, 300, 500), emphasizing the critical role of service performance in ATP Rank outcomes. Return statistics yielded fewer statistically significant results, with Total Points Won emerging as a significant variable in the ATP 100 category. Parameter Estimates revealed five key statistically significant findings, reinforcing the significance of Age, Height, and Total Points Won in shaping player

rankings across various ATP Rank categories. These results indicate that older ATP tennis players are more likely to secure positions among the top 100 ATP players, and height positively influences ranking outcomes. In summary, the MLR analysis elucidated the pivotal roles of age and height in shaping the rankings of professional tennis players. These insights offer actionable information for coaches, players, and stakeholders, facilitating the optimization of training strategies and performance assessment within the professional tennis domain.

Our Neural Network analysis for multiple variables associated with ATP Rank assessment delivered several key insights. Models associated with ATP Rank (500) exhibited superior test validity, with the best model achieving up to 95.1% accuracy, highlighting their predictive power. Furthermore, the first two less accurate models were deemed inadequate on model performance across different outcome categories (according to AUC), while model 3.1 (utilizing all variables as predictors), with 96.8% accuracy (after post-fine-tuning), was identified as the most valid model. In conclusion, Neural Network models have demonstrated the potential to predict outcomes associated with ATP Rank assessment, with ATP Rank (500) models being particularly promising. Further refinement and optimization of these models has the potential to provide valuable insights for analysts and coaches in the tennis domain, aiding their understanding of factors influencing player rankings and improving the accuracy of predictive models.

Overall, our study contributes to a deeper understanding of the factors influencing ATP rankings and provides practical implications for enhancing training strategies and performance assessment in the realm of professional tennis. These findings underscore the significance of specific performance metrics in assessing tennis players' success and predicting their rankings. Coaches, players and stakeholders can use these insights to tailor training strategies and improve their understanding of the key factors influencing competitive performance. Furthermore, ongoing research and refinement of predictive variables have the potential to improve ranking predictions, facilitating more informed decision-making and performance optimization in professional tennis.

Because there are so many possible linear and nonlinear interactions that cannot be effectively monitored, the use of AI in tennis, which is capable of handling these complexities when properly configured and trained, is highly promising. We believe that in the future, it will be utilized more extensively, not only in tennis but also in various other domains where the outcome depends on numerous different variables, all of which are accessible and measurable. A viable solution involves combining multiple AI models that are interconnected. However, it is crucial to emphasize that while AI is a suitable and powerful tool, it always remains just that— a tool to assist decision-making. The ultimate decision should be carefully considered and made by experienced individuals.

## Ethical statement

Given that this research exclusively relied on secondary (freely available) data, formal approval from an ethical review board was not deemed necessary. Nonetheless, ethical considerations remained a top priority throughout the study. The utilization of AI, AI-assisted, and Machine Learning technologies adhered to ethical guidelines and standards. The research design, data collection, and analysis processes were executed with integrity and transparency. Potential conflicts of interest were disclosed, and every effort was made to present the results objectively and without bias.

## Supporting information

**S1 Fig. The Neural Network diagram for 3.1 model.**
(TIF)

## Acknowledgments

In this study, we utilized AI, AI-assisted, and Machine Learning technologies to address the research objectives. We declare that the deployment of these technologies was conducted methodically, under controlled conditions, and adhering to ethical standards, with subsequent result verification by all authors. Our commitment to transparency extends to the assurance that the ethical aspects of our approach were rigorously implemented, and the results were thoroughly scrutinized collectively by the research team.

## Author Contributions

**Conceptualization:** Michal Bozděch.

**Data curation:** Dominik Puda, Pavel Grasgruber.

**Formal analysis:** Michal Bozděch.

**Methodology:** Michal Bozděch.

**Supervision:** Michal Bozděch.

**Validation:** Michal Bozděch.

**Visualization:** Michal Bozděch.

**Writing – original draft:** Michal Bozděch.

**Writing – review & editing:** Dominik Puda, Pavel Grasgruber.

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
