## [Decision Letter · Decision Letter 0]

12 Jun 2024

PONE-D-24-05081A detailed Analysis of Game Statistics of professional tennis players: An Inferential and Machine Learning ApproachPLOS ONE

Dear Dr. Bozděch,

Thank you for submitting your manuscript to PLOS ONE. After careful consideration, we feel that it has merit but does not fully meet PLOS ONE’s publication criteria as it currently stands. Therefore, we invite you to submit a revised version of the manuscript that addresses the points raised during the review process.

**ACADEMIC EDITOR: Please address all the comments and a proof reading is highly recommended.**==============================

We look forward to receiving your revised manuscript.

Kind regards,

Qichun Zhang, PhD

Academic Editor

PLOS ONE

Journal Requirements:

3. Thank you for uploading your study's underlying data set. Unfortunately, the repository you have noted in your Data Availability statement does not qualify as an acceptable data repository according to PLOS's standards.

Additional Editor Comments:

The main issue has been noticed by the reviewers is the writing problem. The manuscript has not been organised well and the contents are not clear. The motivation of the research work is not described well and the results lack of analysis. Thus, a proof reading is needed to clarify the concerns from the reviewers and the innovation should be highlighted clearly in the revised version.

Reviewers' comments:

Reviewer's Responses to Questions

**Comments to the Author**

1. Is the manuscript technically sound, and do the data support the conclusions?

Reviewer #1: Yes

Reviewer #2: No

Reviewer #3: Yes

2. Has the statistical analysis been performed appropriately and rigorously? 

Reviewer #1: Yes

Reviewer #2: No

Reviewer #3: Yes

3. Have the authors made all data underlying the findings in their manuscript fully available?

Reviewer #1: Yes

Reviewer #2: No

Reviewer #3: Yes

4. Is the manuscript presented in an intelligible fashion and written in standard English?

Reviewer #1: Yes

Reviewer #2: No

Reviewer #3: Yes

5. Review Comments to the Author

Reviewer #1: Figure 7 Writing is not clear.

language article clear, correct.

The article is in pdf version

If the word would be better for me to confirm some things, it should be more justified .

In all tables follow the same guidelines for formatting.

Reviewer #2: The abstract needs to be specific on how the data was collected with the design of the study

The introduction needs to be redone. Line 43 is not clear . what does these mean .the introduction has all mixed information which does not clearly show what the authors are focusing at. the aim of research is not clear.

Reviewer #3: I respect the effort been made from the author. The abstract introduces the significance of using Artificial Intelligence, specifically Neural Networks to help ATP ranking with using all factors of ATP's ranking, instead of going through long process. I can see this paper was written well, but the author need to add more information in (Statistical analysis section) in Utilizing the Multivariate Analysis of Variance.

Also, the section(Neural Network Analysis of Multivariate Outcomes Associated

514 with ATP Rank) lacks some details, and need to be more clear.

other than that I can see all other information been added as it should be.

6. PLOS authors have the option to publish the peer review history of their article (what does this mean?). If published, this will include your full peer review and any attached files.

Reviewer #1: **Yes: **ALI ALOUI

Reviewer #2: No

Reviewer #3: No

---

## [Author Response · Author response to Decision Letter 0]

25 Jun 2024

Thank you for your time and feedback. We believe that your comments make the current version of the manuscript much better than the original one.

---

## [Decision Letter · Decision Letter 1]

6 Aug 2024

A detailed Analysis of Game Statistics of professional tennis players: An Inferential and Machine Learning Approach

PONE-D-24-05081R1

Dear Dr. Bozděch,

We’re pleased to inform you that your manuscript has been judged scientifically suitable for publication and will be formally accepted for publication once it meets all outstanding technical requirements.

Kind regards,

Qichun Zhang, PhD

Academic Editor

PLOS ONE

Additional Editor Comments:

2 reviewers out of 3 returned the review reports. Both of them believe that the paper can be published while the quality has been improved after the revision. One reviewer for the original submission has declined the invitation twice, due to the time issue, the decision is made without reviewer 2's comments.

Reviewers' comments:

Reviewer's Responses to Questions

**Comments to the Author**

1. If the authors have adequately addressed your comments raised in a previous round of review and you feel that this manuscript is now acceptable for publication, you may indicate that here to bypass the “Comments to the Author” section, enter your conflict of interest statement in the “Confidential to Editor” section, and submit your "Accept" recommendation.

Reviewer #1: All comments have been addressed

Reviewer #3: All comments have been addressed

2. Is the manuscript technically sound, and do the data support the conclusions?

Reviewer #1: Yes

Reviewer #3: Yes

3. Has the statistical analysis been performed appropriately and rigorously? 

Reviewer #1: Yes

Reviewer #3: Yes

4. Have the authors made all data underlying the findings in their manuscript fully available?

Reviewer #1: Yes

Reviewer #3: Yes

5. Is the manuscript presented in an intelligible fashion and written in standard English?

Reviewer #1: Yes

Reviewer #3: Yes

6. Review Comments to the Author

Reviewer #1: (No Response)

Reviewer #3: All comments that been sent to authors were addressed. Authors did follow all comments. I believe now the manuscript is ready to be published. Moreover, I see that the paper will be helpful in the future as reference.

7. PLOS authors have the option to publish the peer review history of their article (what does this mean?). If published, this will include your full peer review and any attached files.

Reviewer #1: **Yes: **Dr.ALI ALOUI

Reviewer #3: No

---

## [Editor Report · Acceptance letter]

15 Aug 2024

PONE-D-24-05081R1 

PLOS ONE

Dear Dr. Bozděch, 

I'm pleased to inform you that your manuscript has been deemed suitable for publication in PLOS ONE. Congratulations! Your manuscript is now being handed over to our production team.

Kind regards, 

on behalf of

Prof. Qichun Zhang 

Academic Editor

PLOS ONE